# How Do Large Language Models Acquire Complex Knowledge From Text?

## Abstract

Despite the success of pre-training, we are unable to replicate it for continual learning. We don't fully understand how large language models (LLMs) acquire knowledge. We design controlled experiments to measure how they learn complex knowledge in the setting of continued pre-training, probing at two levels of generalization: factual and compositional. First, we show that paraphrasing enables scalable acquisition of knowledge, in which repetition increases learning with diminishing returns. Second, we find that auxiliary views of the underlying knowledge, which formulate and communicate the same knowledge in different ways, yield significantly better generalization. This generalization extends to both compositional knowledge and even factual recall. We postulate that these auxiliary views frequently occur in pre-training corpora and construct a sort of scaffolding. Third, we find that LLMs can possess only a partial understanding of the prior knowledge required for domain adaptation, and bridging these gaps markedly increases learning. Lastly, we examine how learning dynamics differ with model size, post-training, data cleaning, and data replay.

## 1 Introduction

Teaching large language models (LLMs) new knowledge via continued pre-training is surprisingly difficult (Wang et al., 2021; Jang et al., 2021; Hu et al., 2023; Ovadia et al., 2024; Hoffbauer et al., 2024). For instance, a 70B model continually pretrained on Wiki-style documents, despite sophisticated augmentation, only recalls 62.7% of facts (Jiang et al., 2024). Despite open pre-training efforts (OLMo et al., 2024; Grattafiori et al., 2024) that nearly saturate knowledge benchmarks (Joshi et al., 2017; Hendrycks et al., 2020), failed efforts to *reproduce this success* for continual learning reveals a major gap in our understanding of how LLMs learn, especially during pre-training.

The sheer scale of pre-training which obscures the underlying dynamics makes this challenging, and existing studies, varying in corpora, model architectures and sizes, and evaluation, often yield conflicting findings. Even data repetition in pre-training has been well-debated (Lee et al., 2021; Taylor et al., 2022; Hernandez et al., 2022; Xue et al., 2023; Muennighoff et al., 2023). Given evidence that suggests post-training primarily elicits and refines existing capabilities from pre-training (Zhou et al., 2023; Ye et al., 2025; Yue et al., 2025), we find scientific efforts to understand pre-training to be even more relevant. While initial efforts have investigated these dynamics (Allen-Zhu & Li, 2024; Chang et al., 2024), they focus on simple biographical facts. It remains unclear how these dynamics generalize to complex knowledge, and many important questions have yet to be explored.

Motivated by the relevance of domain adaptation (Bai et al., 2025; Sellergren et al., 2025; Wang et al., 2025a; Luo et al., 2025a; Colombo et al., 2024; Singhal et al., 2025), we focus on the setting of learning domain-specific knowledge via continued pre-training. Our study addresses three gaps:

**RQ 1:** What role does paraphrasing play during pre-training? Existing literature offers conflicting views on paraphrasing, and so we revisit this question.

**RQ 2:** How should knowledge be fundamentally formulated textually? Humans are largely affected by how knowledge is presented and it's not clear how these dynamics manifest for language models.

**RQ 3:** What are the unique challenges in continued pre-training? Specifically, does the prior knowledge gap in LLMs' existing knowledge bottleneck adaptation? Moreover, does knowledge injected during continued pre-training manifest immediately or emerge via post-training?

To investigate these questions, we design a controlled setup using recent arXiv papers to proxy self-contained knowledge domains. We continue pre-training on the papers and related reformulations, in order to study the acquisition of *complex knowledge*, building upon the LLM's existing knowledge. We build our evaluation framework using a bottom-up approach, mapping each knowledge-bearing sentence to factual probes in order to study if the LLM remembers *atomic facts* and constructing compositional probes to study if the LLM learns to *piece knowledge together*. This setup allows us to isolate confounding factors to the best of our abilities and suggest causal linkage.

Our experiments reveal three main findings: (1) Acquisition of new knowledge requires repetition and paraphrasing enables repetition without overfitting; however, with sufficient repetition, we observe grokking that occurs across both settings (2) Auxiliary views dramatically improve the learning of both compositional and even factual knowledge (3) Bridging knowledge gaps by first training the LLM on prerequisite concepts improves learning. These results lead us to a central conjecture: LLMs largely lack the ability to synthesize complex, novel information from primary sources alone. Instead, their learning relies on *auxiliary views*, a web of explanations, analogies, and reformulations that humans generate as they learn, process, and teach each other. We posit that this enables a generalizable encoding of knowledge akin to scaffolding, suggesting that pre-training's success is not merely due to scale.

To our knowledge, this is the first study to isolate and examine the dynamics of complex knowledge acquisition at the data level, and we also investigate questions specific to continued pre-training that have been overlooked in prior work. We hope these findings help provide a causal lens on pre-training and, along with our framework, catalyze further scientific inquiry into LLMs.

## 2 RELATED WORKS

**LLMs and Data.** Model size and data volume are the primary determinants of LLMs' capabilities (Brown et al., 2020; Kaplan et al., 2020; Carlini et al.; Tirumala et al., 2022). While model size is usually emphasized, Hoffmann et al. (2022) shows optimal performance requires scaling data together. Further, Kandpal et al. (2023) causally links QA performance to the number of relevant pre-training documents. Data composition is also important (Albalak et al., 2024), including factors like quality (Gunasekar et al., 2023; Longpre et al., 2024), diversity (Chen et al., 2025), deduplication (Lee et al., 2021; Zhang et al., 2022; Raffel et al., 2020), and filtering (Weber et al., 2024; Li et al., 2024). For instance, Dodge et al. (2021) analyzes C4 and reports an unexpected concentration of U.S. patents and government websites. Such skews are misaligned, motivating continued pre-training on relevant sources like arXiv (OLMo et al., 2024). However, these findings focus on corpus characteristics, overlooking fundamental questions of how knowledge should be represented.

**Controlled Studies of Pre-training.** Notable efforts investigate how pre-training instills knowledge. Allen-Zhu & Li (2024) show that paraphrased augmentation increases the memorization of biographical facts from 9.7% to 96.6% . They further find that incorporating QAs about facts during pre-training leads to increased encoding on even held-out facts, whereas inserting QAs during instruction-tuning does not. This highlights how data formulation during pre-training strongly affects knowledge encoding and generalization. Meanwhile, Chang et al. (2024) intermittently injects fictional facts during pre-training, observing that knowledge is acquired incrementally upon each exposure and is then subject to decay, suggesting a gradual, not emergent, learning process. They also provide opposing views on the benefits of paraphrasing. We build upon these works to study how LLMs learn complex knowledge during continued pre-training. Furthermore, we offer a causal explanation for these reported differences on paraphrasing.

**Domain Adaptation via Continued pre-training.** Continued pre-training is being adopted in varying levels for domain adaptation. For instance, MedGemma (Sellergren et al., 2025) and TxGemma (Wang et al., 2025a) perform zero text-based continued pre-training and only multi-task finetuning, while Intern-S1 (Bai et al., 2025) does science-specific mid-training for 5 trillion tokens and outperforms frontier LLMs on science benchmarks. This variance is likely due to the difficulties with continued pre-training (Wang et al., 2021; Jang et al., 2021; Hu et al., 2023; Ovadia et al., 2024; Hoffbauer et al., 2024; Jiang et al., 2024). Since existing studies focus on catastrophic forgetting (Luo et al., 2025b; Bethune et al., 2025) and stability (Guo et al., 2024; Gupta et al., 2023; Ibrahim et al., 2024), we uniquely examine the opportunity cost of data replay, investigating the extent to

which preserving prior capabilities compromises domain adaptation. We also investigate whether bridging the prior knowledge gap in LLMs is a necessary condition for learning new knowledge.

We acknowledge other forms of knowledge injection (Ovadia et al., 2024; Song et al., 2025). Retrieval augmentation helps with factual knowledge, and domain-specific tasks can be addressed by fine-tuning or in-context learning. However, we motivate our study due to evidence that pre-training is the primary driver of LLMs' intelligence (Zhou et al., 2023; Ye et al., 2025; Yue et al., 2025).

## 3 EXPERIMENTAL SETUP

**Problem Formulation.** The pre-training objective for an autoregressive LM, $f$ parameterized by $\theta$, is next-token prediction. Given a corpus $C$ composed of documents $d_1, \ldots, d_M$, each a sequence of tokens $(t_{m,1}, \ldots, t_{m,n_m})$, the model is trained to maximize the log-likelihood:

$$L(\theta) = \sum_{m=1}^{M} \sum_{i=1}^{n_m} \log P(t_{m,i} \mid t_{m,<i}; \theta) \tag{1}$$

Unlike discriminative tasks with low-dimensional label spaces, next-token prediction requires the model to predict within a high-dimensional vocabulary. To optimize this objective across a vast corpus, $f_\theta$ internalizes grammar, knowledge, and reasoning patterns to anticipate each token $t$. Consequently, the serialization and composition of these tokens, as words, sentences, and documents, are non-trivial. Further, we denote domain knowledge abstractly as $K$, regardless of its underlying structure, which is accessible through observable documents in corpus $\mathcal{C}_K = \{ d_i \in \mathcal{C} \mid d_i \text{ contains information relevant to } K \}$. This leads to our central research question: How should knowledge be presented as text? What are the properties of $d \in \mathcal{C}_K$ that help $f_\theta$ acquire $K$?

**Observation**. Knowledge is rarely represented once in a corpus. Consider the paper, "Attention Is All You Need" (Vaswani et al., 2017). This primary source is followed by a proliferation of documents that represent similar, underlying knowledge: tutorials on how to code it, blog posts offering intuitive explanations, forum discussions clarifying ambiguities, and textbooks situating it within broader ideas in a pedagogical manner. It is a *natural byproduct of society's collective effort to learn, process, and teach knowledge*.

We term these textual manifestations of the same underlying knowledge **auxiliary views**. We make this distinction because human-generated views are rarely mere paraphrases, instead discussing the knowledge in diverse contexts (e.g., Transformers' comparative advantages against different architectures), distinct forms (e.g. blogs or forums), and often with a pedagogical lens. These views collectively provide a more *complete, contextualized* picture of the knowledge. We posit that pre-training corpora are replete with these views, potentially explaining their effectiveness. In our study, we frame paraphrasing as linguistic variations of a singular view and set it as a control.

As we investigate this question, we extend our inquiry to three critical aspects of domain adaptation. First, we revisit the effectiveness of paraphrasing. Second, we examine prerequisite knowledge, asking whether pre-training corpora naturally bridge the gap to target domains or if explicit gap-filling is necessary. Finally, we analyze the learning versus forgetting trade-off, specifically the cost of data replay on learning and its subtle impact on knowledge emergence during post-training.

### 3.1 DATASET

**Domains**. We use six Computer Science arXiv papers to represent distinct domains, as method papers exemplify encapsulated knowledge. To mitigate leakage, we select papers published after the OLMo2 (OLMo et al., 2024) cutoff and manually verify their absence in the pre-training corpus. Our pre-processing pipeline follows Lewkowycz et al. (2022) with additional steps such as resolving LaTeX macros. For each domain, we generate paraphrases, prior knowledge, and auxiliary views for each paper in the dataset, utilizing GPT-4.1 for paraphrasing and GPT-5 for the rest. Inspired by prior works (Gunasekar et al., 2023; Allen-Zhu & Li, 2024; Jiang et al., 2024), we focus on textbooks, Stack Exchange–style Q&A, and blogs for auxiliary views. Prompts are detailed in Appendix A.14.

Table 1: The *factual* probe extracts questions from knowledge-bearing sentences, converts them into statements with answers at the end, and adds context. Notice how the italicized words in the original sentence reappear. The *compositional* probe composes knowledge from one or multiple sentences (see related sentence) to infer information that was not explicitly stated before. In this case, it makes a 1-hop inference presuming optimization knowledge, with regards to $\beta$ if it becomes large. The **target span** of each probe is bolded.

| | |
|---|---|
| **Original sentence** | ...The *added constraint is important*, as it *prevents the model from* **deviating too far** from the distribution on which the reward model is accurate, as well as maintaining the generation diversity and preventing mode-collapse to single high-reward answers... |
| **Factual probe** | In the paper 'Direct Preference Optimization: Your Language Model is Secretly a Reward Model', the authors state that the *added constraint* in the RL fine-tuning objective, the KL divergence term $D_{\mathrm{KL}}(\pi \| \pi_{\mathrm{ref}})$ associated with the base reference policy $\pi_{\mathrm{ref}}$ (the SFT model), *is important because it prevents the language model policy from* **deviating too far**. |
| **Related sentence** | ...where $\beta$ is a parameter controlling the deviation from the base reference policy $\pi_{\mathrm{ref}}$, namely the initial SFT model $\pi^{\mathrm{SFT}}$ (...) The *added constraint is important*... |
| **Compositional probe** | According to the paper 'Direct Preference Optimization: Your Language Model is Secretly a Reward Model', if the learned reward model were perfectly accurate but the KL penalty $\beta$ became extremely large, the optimized policy $\pi_\theta$ would converge toward the **reference policy**. |

**Probes**. To measure learning, we adopt LLaMa-style probes (Petroni et al., 2019; Jiang et al., 2020; Zhong et al., 2021) where the task is to predict the last word in the sentence. Like Chang et al. (2024), we relax this to multiple words, i.e. the target span. In our study, we design two types of probes: (1) Factual probes measure the recall of explicit information stated in the text (2) Compositional probes require building upon one or multiple facts to infer information not explicitly state (Table 1). We carefully design a pipeline to construct them. For factual probes, we first filter for knowledge-bearing sentences within the papers. From each, we generate QA pairs and convert them into contextualized, self-contained statements, in which the answer appears at the end. We automate these steps with GPT-5 (OpenAI, 2025), and verify that GPT-5 sufficiently understands the papers. Our process yields 2027 factual and 271 compositional probes, a scale exceeding Chang et al. (2024). We manually validate 100 factual and all compositional probes to ensure their quality. Full details of the pipeline are available in Appendix A.13. The dataset and code is publicly available.[1]

## 3.2 TRAINING

We use the OLMo-2 base models (1B, 7B, 13B, and 32B). Following Chang et al. (2024), we employ a single-batch knowledge injection strategy: documents for each domain fit in one forward pass, The rest of the batch is filled with data replay. We perform $N = 100$ knowledge injections for most experiments. For *Source*, the original paper is injected in every batch. For *Para. 9*, we cycle through the original document followed by nine paraphrased versions, cycling $\frac{N}{10}$ times. For *Para. 9 + Aux. Views*, auxiliary views are injected alongside the papers, replacing data replay, and follow a separate cycle. For textbooks, as an example, we inject one chapter per batch. We also test interleaving additional batches of data replay in between injections. We use a cosine warm-up and decay LR schedule (Ibrahim et al., 2024). Full hyperparameters are detailed in Appendix A.12.

**Data Replay.** We stream examples from the DCLM subset (Li et al., 2024) of OLMo2's pre-training corpus for data replay. We ensure each replayed token is continuous from the previous fetch.

**Metrics.** We evaluate the model's performance on our probes after each training step:

- **Log-Probability**: We measure the average log-probability assigned to the ground-truth tokens of the target span. Greedy decoding can be unstable for multi-token target spans. To evaluate each token prediction accurately, we provide the preceding ground-truth token as context, a teacher-forcing approach also adopted by Chang et al. (2024).

- **Hits@k**: Following (Petroni et al., 2019), this metric is 1 if the correct token is present in the model's top-k predictions. For multi-token target spans, we report the mean.

To quantify the lexical bias potentially introduced by paraphrases and auxiliary views, we also propose an adaptation of the BM25+ metric (Trotman et al., 2014), which we term Source-Weighted BM25+ (see Appendix A.3). We leverage this when controlling for the leakage of our probes.

---

[1]https://anonymous.4open.science/r/fine-tuning-or-retrieval-D779/.

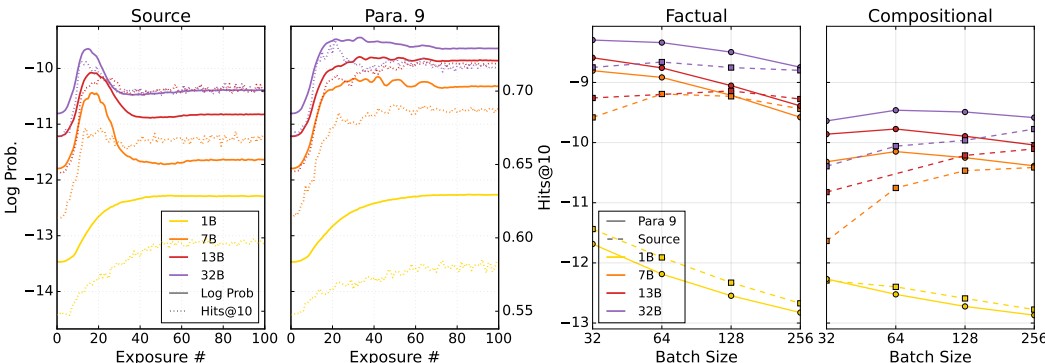

Figure 1: **(Left)** Training OLMo2-7B with a batch size of 64 on *Source* versus *Para. 9*, tracking performance on compositional probes on log probabilities and hits@10. **(Right)** Log probabilities of probes at the end of individual training runs across model sizes and batch sizes.

## 4 PARAPHRASING HELP MODELS LEARN BETTER

In the first panel of Figure 1, we see that repeated exposure initially improves compositional generalization until ∼20 exposures, after which performance saturates and sharply decreases before stabilizing. This degeneration is expected as the model repeatedly trains on identical texts and likely overfits. By contrast, *Para. 9* **prevents this collapse** and allows continued improvement until ∼40 exposures. The same trend holds for factual probes (Figure 8). Notably, these benefits only emerge in model sizes 7B and larger; the 1B model yields slightly worse results with paraphrasing, likely not reaching its overfitting regime under our experiment's batch-size and data-mix configuration.

These dynamics are sensitive to learning rate and batch size. Higher learning rates exacerbates and accelerates the collapse in *Source* (Figure 20), and even *Para. 9* degrades at higher learning rates although further paraphrasing mitigates this. Increasing batch size, thereby increasing data replay, has the opposite effect: in the rightmost panel of Fig 1, *Source* becomes increasingly stable and approaches *Para. 9* at batch size 256, while *Para. 9* stays roughly constant, indicating that paraphrasing's compositional gains primarily arise from preventing degeneration that larger batches already do. We find that larger batch sizes are not the causal factor alone, as a similar effect can be achieved with interleaving steps of data replay with a batch size 32 (Figure 5), although this induces lower learning in comparison due to forgetting during unrelated training steps (Chang et al., 2024).

Beyond collapse prevention, paraphrasing uniquely enhances the **semantic acquisition of factual knowledge**. Larger batches yield almost no factual improvement—except a small gain at batch size 64 for 7B—whereas paraphrasing consistently helps models 7B and above. Although factual probes are extracted from explicit sentences, they are contextualized to become atomic statements and thus *Source*'s verbatim memorization generalizes worse, while paraphrases expose the model to semantically varied formulations. However, the paraphrasing advantage shrinks at larger batch sizes and reverses slightly at batch size 256 (7B, 13B), likely due to increased replay diluting gradients from paraphrased texts. We believe this explains conflicting prior work: Chang et al. (2024), using a batch size of 2048 with 2048-token chunks, reports degraded factual learning with paraphrasing, while Allen-Zhu & Li (2024), using batch size 96 with 512-token chunks, reports large gains.

Because log-probabilities provide limited insight, we also examine Hits@k metrics. In Figure 9, compositional performance under *Source* is stable at Hits@1 and Hits@10, with paraphrasing gains emerging only at Hits@100. This suggests that paraphrasing primarily improves distributional confidence for the hard probes, likely corresponding to one-off facts rather than repeatedly stated points. Furthermore, both **factual and compositional probes remain difficult** even at 32B (Figures 6, 7). After filtering probes whose targets don't appear in the text, performance still plateaus (e.g., 78% at Hits@10). This aligns with Jiang et al. (2024) and Ovadia et al. (2024), which similarly find that factual knowledge is hard to instill via continued pre-training. Since facts are seen at least a hundred times and learning saturates by ∼40 exposures, why certain facts remain unlearned is unclear.

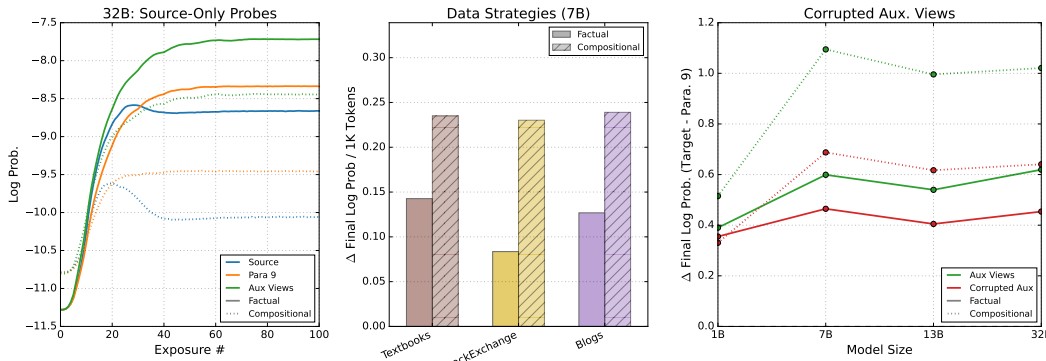

Figure 2: **(1)** Effect of auxiliary views on compositional probe performance. Training with auxiliary views (grey shaded region) yields a sharp post-training improvement when models return to the original and paraphrased documents. **(2)** Auxiliary views also lead to better generalization on the factual knowledge.

## 5 LARGE LANGUAGE MODELS NEED SCAFFOLDING TO LEARN

First, we observe that incorporating auxiliary views substantially enhances performance on both factual and compositional probes. Despite the fact that the original papers were theoretically sufficient, the addition of auxiliary views appears to facilitate the LLM's acquisition of a deeper, more generalizable encoding of the knowledge presented in the paper, similar to the concept of scaffolding (Hammond, 2001).

To address the potential confounding factor of lexical bias—where the model might favor specific phrasing because both the auxiliary views and the probes were generated by models from the same family (GPT-5-mini and GPT-5, respectively)—we analyzed performance across stratified probe sets (Figures 10, 11, and 12). While a lexical advantage is observable for probes whose target answers appear exclusively in the auxiliary views, the performance gains are demonstrably not confined to this set. Crucially, improvements extend to probes with targets present in both the auxiliary and original documents, as well as to those with targets found only in the original documents. This distribution of gains validates a genuine generalization of knowledge rather than surface-level pattern matching or answer leakage. Furthermore, to ensure that the entire probe, and not just the answer, does not introduce a lexical bias, we analyzed the overlap of probe words across the texts using our Source-Weighted BM25+ metric. We observe improvement across most probes, regardless of positive or negative overlap (Figure 14).

To stress test this concern further, we performed an ablation study where we explicitly removed all non-stop words from the auxiliary views that also appeared in the probes. The right side of Figure 2 shows that while performance is lower than using the full auxiliary views, it still yields significant increases compared to the baseline paraphrased condition. This final experiment confirms a clear benefit derived from the knowledge introduced by the auxiliary views.

Furthermore, the efficacy of this approach also *emerges with model size*. As shown in Figure 10, the 1B model does not benefit from auxiliary views in the same way; while it shows strong performance on lexically-aligned probes, this advantage fails to generalize for lexically non-aligned probes and performs worse. Furthermore, we repeat our experiments with human-generated auxiliary views found from the open web and we also find the helpfuless of these texts grow with model size (Figure 13), confirming that this is not an artifact of our synthetic text.

Unexpectedly, auxiliary views also improve performance on factual probes. This is counterintuitive, as these probes are direct paraphrases of the source material, and diverting training time to auxiliary content could be presumed to hinder direct memorization. We hypothesize that the **auxiliary views help the model build a more structured knowledge representation or 'schema,'** (Spiro, 2017) which facilitates more effective recall of facts. Furthermore, an ablation study on the components of the auxiliary views reveals that each of the texts are similarly effective, indicating a *synergistic effect* where multiple, diverse views contribute to a more comprehensive understanding.

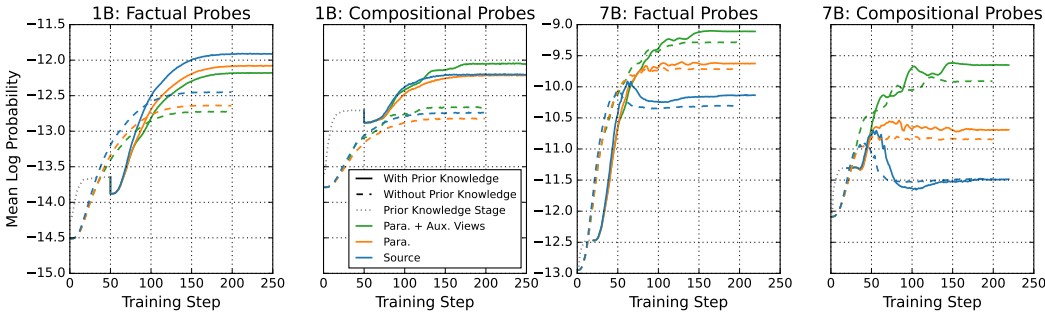

Figure 3: **(1)** An initial stage of prior knowledge learning increases learning on 1B and 7B models across all three models. **(2)** Probes are implicitly learned even before seeing the documents simply by increasing foundational knowledge.

# 6    PRIOR KNOWLEDGE MATTERS

Table 2: Prior Knowledge Gap

| Model | Base | CPT |
|---|---|---|
| OLMo-2-0425-1B | 0.4380 | 0.5454 |
| OLMo-2-1124-7B | 0.6859 | 0.7272 |

While recent studies have begun to explore this question (Wang et al., 2025b; Gekhman et al., 2024; Yang et al., 2024), this body of work remains nascent, motivating our investigation. We first establish the existence of a "prior knowledge gap" by generating Multiple-Choice Question-Answer (MCQA) pairs on prerequisite topics for each of the papers. As shown in Table 2, initial model performance is low, but improves after training on synthetically generated textbooks that cover this foundational knowledge. Crucially, Figure 3 demonstrates that continued pre-training on models imbued with this prior knowledge enhances downstream performance across all three of our methods, signifying the importance of addressing such deficits in practical domain adaptation. Moreover, the prior-knowledge training stage itself improves probe performance, indicating that foundational knowledge fosters generalizable understanding that transfers over. Additionally, we find that prior knowledge does not account for the success of auxiliary views. The performance of models trained with auxiliary views, even here, suggests that these broader perspectives are essential beyond merely aiding contextualization, and in fact outperform counterparts more than before in Figure 1.

# 7    ADDITIONAL ABLATIONS

**Data Replay Hurts Learning the Target Domain.** Data replay is widely studied as a strategy to mitigate catastrophic forgetting. To our knowledge, its impact on acquiring the target domain of continued pre-training has not been examined. While this question can be less impactful and perhaps obvious, we want to portray the exact tradeoffs being made when data replay is introduced so that practitioners can make a mindful decision. Furthermore, many researchers are using LLMs for uses cases such as, as single-task classifiers, that care less for its general capabilities on unrelated domains.

Figure 5 demonstrates a significant benefit to compositional knowledge from minimal data replay, which can mitigate the degradation we observed from earlier without substantially impeding factual learning. However, the efficacy is highly dependent on the integration method. Specifically, we observe that interleaving data replay steps between subsequent exposures likely induces gradual forgetting of the desired knowledge (Chang et al., 2024) and ultimately leads to reduced overall learning.

**Interactions with Post-Training** A common step after continued pre-training is task-specific fine-tuning or instruction tuning. Furthermore, prior suggest that this is where pretrained knowledge

emerges (Zhou et al., 2023). Further, Sun & Dredze (2025) suggests that knowledge instilled during continued-pretraining only emerges after fine-tuning. We study whether (a) the pretrained knowledge is forgotten or (b) emerges during this process. Figures 15 and 16 reveal a consistent pattern: models that have less replay often recover much of their performance and surpass some of their counterparts, in which we observe a novel interaction between data replay and emergence of knowledge. We leave this as an open question for future work.

**Training on Long Documents.** One challenge in training on domain knowledge is that it typically cannot be contained within a single context window, as is the case in our dataset. To address this, we evaluate overlapping chunks but observe only marginal benefits, as shown in Figure 18.

## 8 DISCUSSION

A central insight of our work is that auxiliary views, such as textbooks, blogs, or Q&A formats, play a role analogous to scaffolding in human learning. Much as young children, and even adults, benefit from well-structured explanations to internalise new knowledge, LLMs appear to benefit substantially when knowledge is presented through re-communicated or explanatory formats. One possible interpretation is Vygotsky's theory of zone of proximal development (Cole et al., 1978), which suggests that learning is bounded by an individual's developmental state and can only progress with the guidance of a more knowledgeable other. Similarly, LLMs appear to be constrained by their current representational capacity and require pedagogical support to generalize new knowledge effectively. While it remains unclear whether larger, more extensively trained models rely *less* on such scaffolding, we argue that the prevalence of auxiliary views in pre-training corpora has been a key factor in enabling LLMs to reach their present level of knowledge. In one way, this is unsurprising as humans suffer from the same problem in which we need help with each other across various domains and it's rather the human as a collective that has a full mastery not any individual.

From a practical perspective regarding domain adaptation, our findings underscore the importance of careful data curation and, in some cases, targeted synthetic augmentation. For broad or foundational knowledge, pre-training corpora already contain abundant auxiliary views, particularly textbooks, which we find to be especially effective. In contrast, open scientific corpora rely heavily on primary research papers that lack such pedagogical scaffolding. Our results suggest that in these settings, augmenting training data with auxiliary views could be crucial for enabling models to acquire knowledge more effectively. Finding such data or synthetically generating these views for the latest scientific knowledge, however, may be a challenge.

**Limitations.** Despite constructing a large number of probes, our analysis is restricted to a small set of computer science papers, which may introduce corpus-level biases. To mitigate this, we demonstrate that our results hold consistently across all six papers in our dataset, as shown in Figure 19. However, computer science is a domain of formal knowledge where ground truth is stable and compositionality is well-defined. It is this controlled setting that allows us to precisely measure the impact of auxiliary views and knowledge gaps on learning dynamics. We acknowledge that these findings may not directly transfer to domains of empirical knowledge (like medicine, which evolves with new evidence) or interpretive knowledge (like law or literary criticism, where 'truth' is a function of argumentation, the governing legal system, and nuance). We believe this is a critical and starkly different challenge that warrants its own investigation, building upon the baseline this work provides.

Our setup also introduces a data scale limitation: our models are only exposed to a handful of new knowledge units at once, whereas pre-training usually involves simultaneously learning thousands of concepts with complex correlations. While we believe our insights should scale under approximate independence between domains, it is possible that emergent capacity limits arise when learning too much knowledge simultaneously. Conversely, large-scale learning might also give rise to unexpected synergies. Studying these effects at scale remains an important open question beyond the scope of this study. Furthermore, while our results carry important implications for understanding the nature of intelligence in LLMs, they are constrained by the scale of models we study (up to 32B), repeatedly emphasized in the literature as the primary determinant of knowledge acquisition in LLMs.

**Future work.** Several extensions follow naturally. First, our results suggest the need to study larger models, which may have sufficient representational capacity to learn beyond the constraints we ob-

serve at the 32B scale. Finally, cross-domain text would help assess whether auxiliary scaffolding is a general phenomenon of LLMs or a domain-specific effect. Furthermore, Allen-Zhu & Li (2023) observes that manipulation of knowledge (e.g. inference) actually requires chain of thought to compose existing knowledge together. Our compositonal probes rely on single-token decoding, and there is much to be studied regarding how pretrained knowledge interacts with inference time reasoning.

# 9 CONCLUSION

The findings of this research yield two primary contributions. First, we present what we believe to be the first to identify (a) the causal significance of auxiliary views, and (b) the distinct challenges associated with continued pre-training, notably including prior knowledge gaps, data replay, and post-instruction emergence. Addressing our core research question, we establish causal links between auxiliary view and enhanced knowledge encoding. These auxiliary views provide a robust explanation for the effectiveness of pre-training corpora beyond simple scale, suggesting that pedagogically structured texts are a critical determinant of successful knowledge acquisition and improved compositional reasoning. Further, our framework offers a robust pipeline for generating probes to extend this line of inquiry into other domains. Second, we offer practical guidelines for practitioners engaged in domain adaptation. Our recommendations include the strategic continuation of pre-training for foundational knowledge, in-batch strategy of data replay, and employing text paraphrasing techniques when data resources are scarce. In conclusion, our evidence underscores the crucial role of auxiliary views in fostering a more generalizable knowledge representation, leading to improvements in both factual recall and compositional reasoning. While acknowledging that the scope and scale of this study may limit full generalizability, we are confident that our hypothesis and supporting results will provoke deeper discussion and stimulate subsequent experimentation into the fundamental mechanisms of knowledge acquisition and intelligence within LLMs. Ultimately, we hope these insights prove valuable to practitioners in the domain adaptation space, encouraging a greater consideration for the importance of data.

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

# A APPENDIX

## A.1 PROBES GENERATION

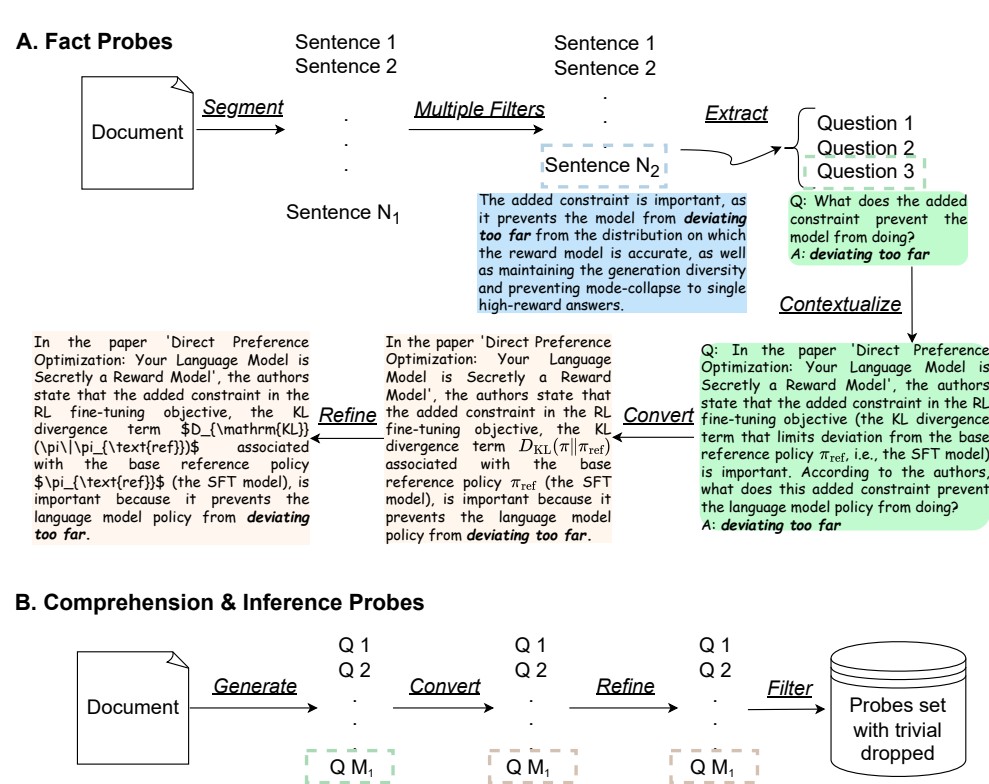

Figure 4: Overview of the Probe Generation Pipelines. **(A) Factual Probe Generation**: We generate probes on a sentence-by-sentence basis to adhere closely to the source document for ground truth. Preprocessing: We first filter sentences that contain minimal knowledge (to prevent noise from structural comments, e.g., "Let us first discuss the following results"). We then use heuristics to filter out sentences that may yield low-quality probes, such as those that are too short (e.g., "the sweep has 22 runs") or contain excessive LaTeX code with no valid English targets for extraction. Question Extraction: From each filtered sentence, we extract 1–3 questions that capture the knowledge contained within. Contextualization: Questions are contextualized to be clear and self-contained, accurately reflecting the knowledge being tested. Cloze Conversion: The questions are converted into cloze statements, placing the answers at the end. Refinement: We refine the statements, ensuring all mathematical content is written in LaTeX and verifying the correctness of the preceding steps. **(B) Compositional Probe Generation**: Recognizing that not all atomic facts necessitate a compositional probe (some only concern simple experimental details), we employ a two-level approach. First, we break the paper into sections and prompt a Large Language Model (LLM) to extract compositional questions. Compositionality is defined as either (1) **Inference** (using reasoning upon supporting text to reach a new insight) or (2) **Synthesis** (combining knowledge across several facts into a synthesized statement). This is performed section-by-section to ensure granularity. Second, we repeat this step by providing the entire paper to the LLM to allow for more holistic questions. Cloze Conversion: Similar to the factual process, these questions are converted into cloze statements with the answer at the end. Refinement: Statements are refined to ensure mathematical content is in LaTeX and to enforce proper self-containment by referencing the paper (e.g., "according to the paper"). Filtering: A comprehensive filtering step is conducted, where we ask the LLM to assess if the questions are too simple, imprecise, or confusing. This step, on average, removes 15% of probes.

## A.2 DATA MATERIALS

Table 3: Data Summary

| Material | Avg. Length (tokens) |
|----------|----------------------|
| Papers | 10472.8 |
| Paraphrased Papers | 10571.7 |
| Blogs | 17858.7 |
| Stack Exchange | 27331.3 |
| Textbooks | 36407.8 |

## A.3 SOURCE-WEIGHTED BM25+

To quantify the lexical advantage present in the synthetic views (e.g., paraphrases, textbooks) we generate over the *Source*, we propose a **Source-Weighted BM25+** metric. Traditional IDF weighting is unsuitable here, as our goal is not to rank documents but to measure the lexical advantage over *Source*. Instead, we substitute IDF with a weight $w_{\text{source}}(t)$ derived from the **Source** term frequency ($\text{tf}_{\text{source}}$). This prioritizes terms that are rare or absent in the original papers but frequent in the synthetic views, effectively capturing the lexical "leakage" of our probes.

For a probe $P$ consisting of $n$-gram terms $t \in P$, and an auxiliary training corpus $C$, the transfer relevance score is defined as:

$$\text{Score}(P, C) = \sum_{t \in P} w_{\text{source}}(t) \cdot \left( \frac{\text{tf}_C(t) \cdot (k_1 + 1)}{\text{tf}_C(t) + k_1 \cdot \left(1 - b + b \cdot \frac{|D_C|}{\text{avgdl}_C}\right)} + \delta \right) \tag{2}$$

where $\text{tf}_C(t)$ is the term frequency of $t$ in the auxiliary corpus $C$, and $k_1$, $b$, and $\delta$ are standard BM25+ hyperparameters (set to $k_1 = 10.0$, $b = 0.75$, $\delta = 1.0$).

$$w_{\text{source}}(t) = \frac{5}{1 + \text{tf}_{\text{source}}(t)} \tag{3}$$

This weighting scheme ensures that terms which are common in the Source contribute negligibly to the score, while terms that are rare in the Source are weighted highly.

## A.4 DATA REPLAY

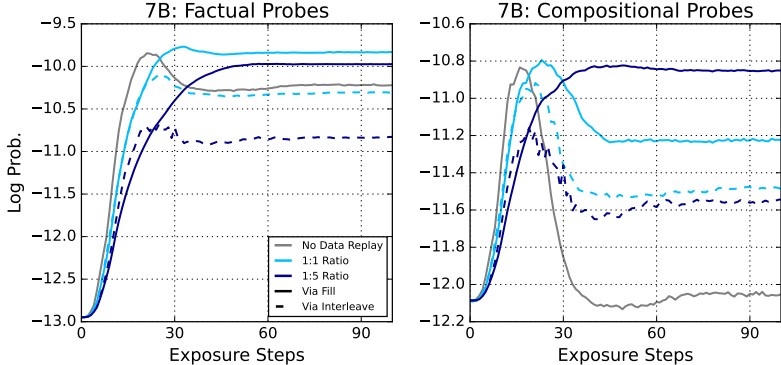

Figure 5: Comparison of factual-probe performance when training OLMo2-7B on *Source* versus *Para. 9* with a batch size of 32. Paraphrasing yields modest improvements when contrasted with the larger gains observed on compositional probes.

## A.5 HITS@K

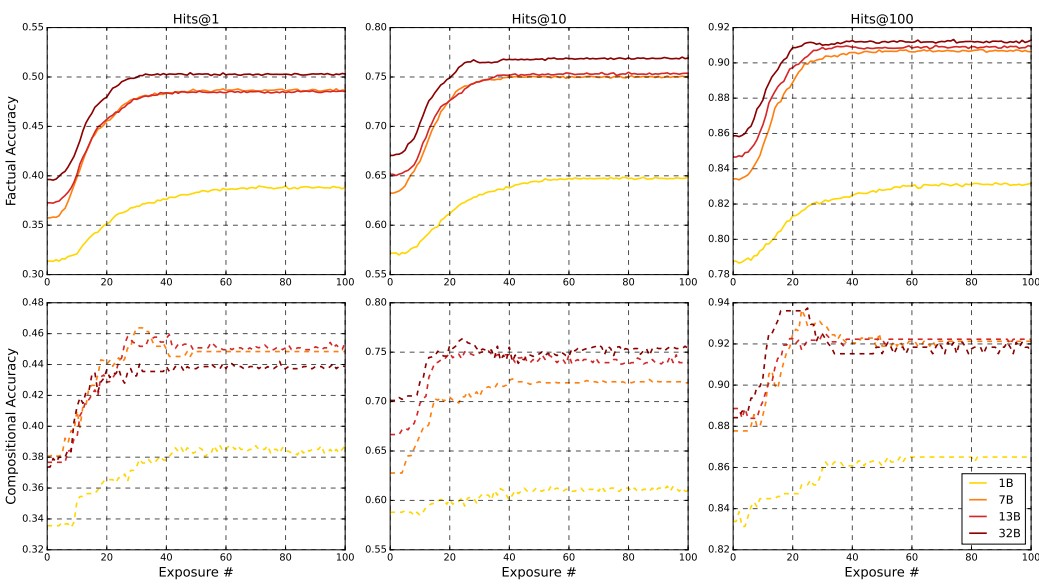

Figure 6: Hits@k progression during training on *Source* across model sizes. For clarity, we restrict the figure to target spans that appear in *Source*, removing any probes absent from the training distribution to avoid confounding effects from inherently difficult or unseen items. Despite extensive training, it is difficult to do well on these probes.

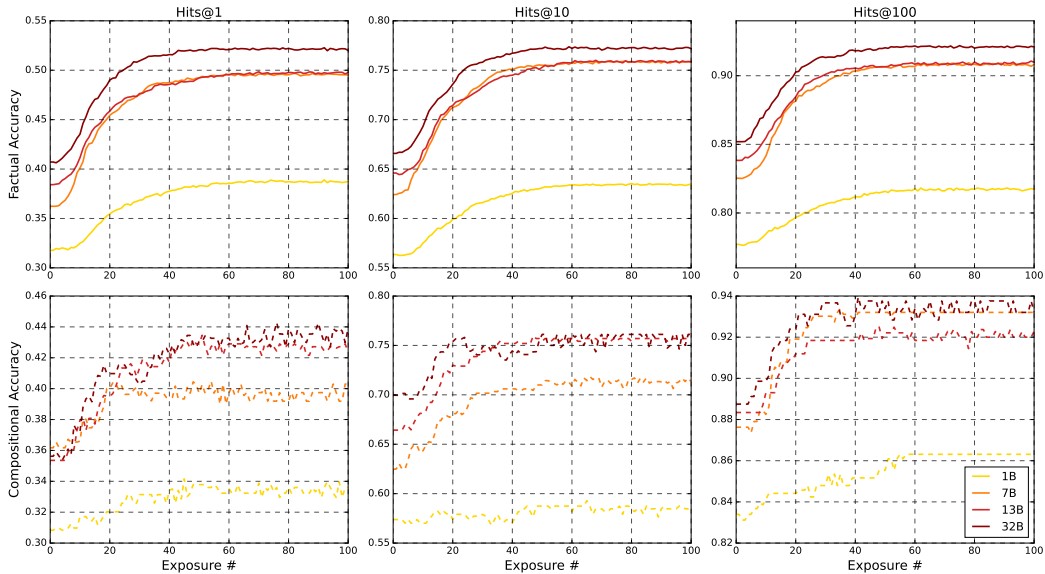

Figure 7: Hits@k progression during training on *Para. 9* across model sizes with a batch size of 64. For clarity, we restrict the figure to target spans that appear in *Para. 9*, removing any probes absent from the training distribution to avoid confounding effects from inherently difficult or unseen items. Despite extensive training, it is difficult to do well on these probes.

## A.6 Additional Plots for Paraphrasing and Repeating Data

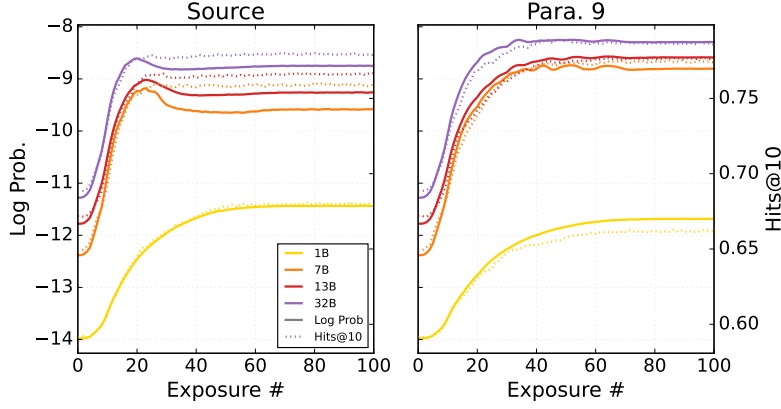

Figure 8: Comparison of factual-probe performance when training OLMo2-7B on *Source* versus *Para. 9* with a batch size of 32. Paraphrasing yields modest improvements on factual probes when contrasted with the larger gains observed on compositional probes.

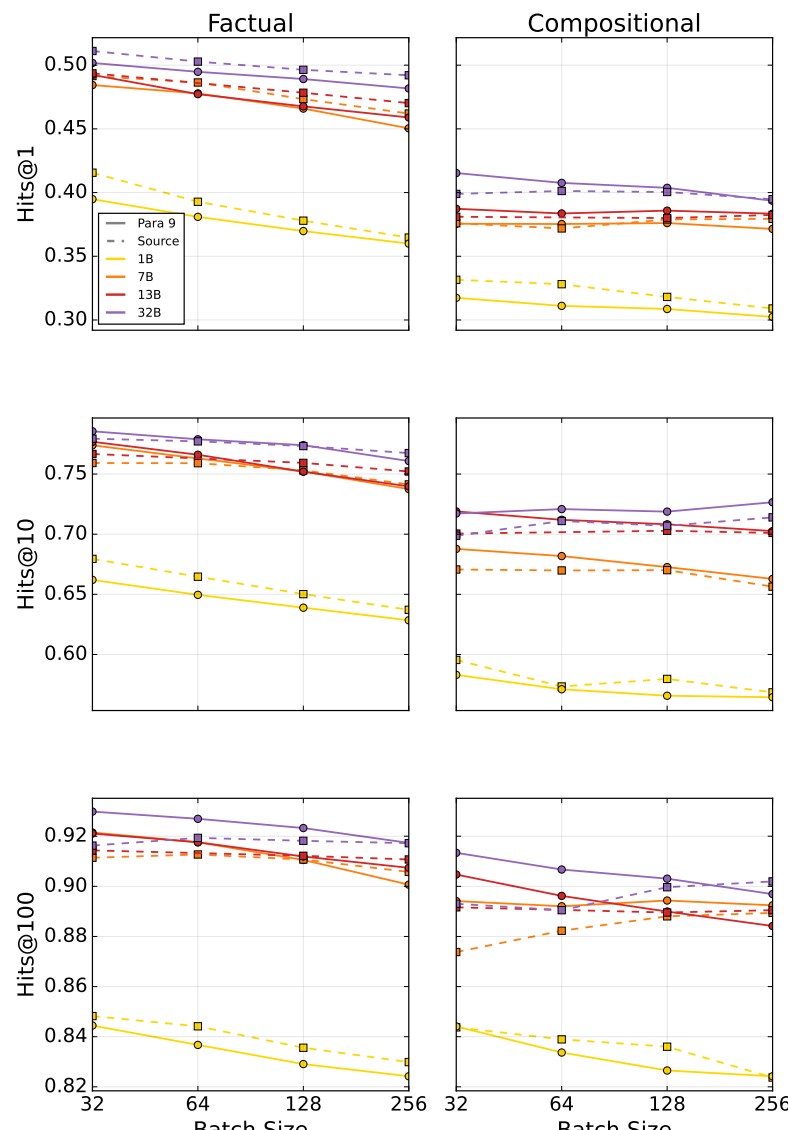

Figure 9: Training on *Source* for 500 exposures with cosine warmup and no decay. We observe an early dip followed by steady increases in target-span log-probabilities. Hits@10 improves as well, though the gains are comparatively modest and experience no initial dip.

## A.7 ADDITIONAL PLOTS FOR AUXILIARY VIEWS

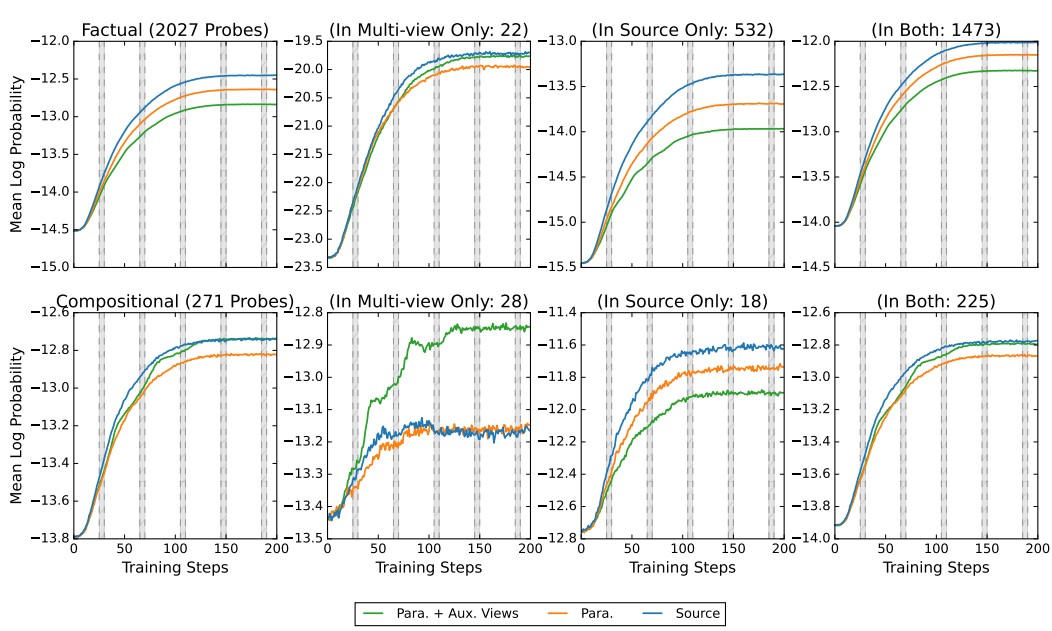

Figure 10: Stratified Performance of 1B across the three data strategies.

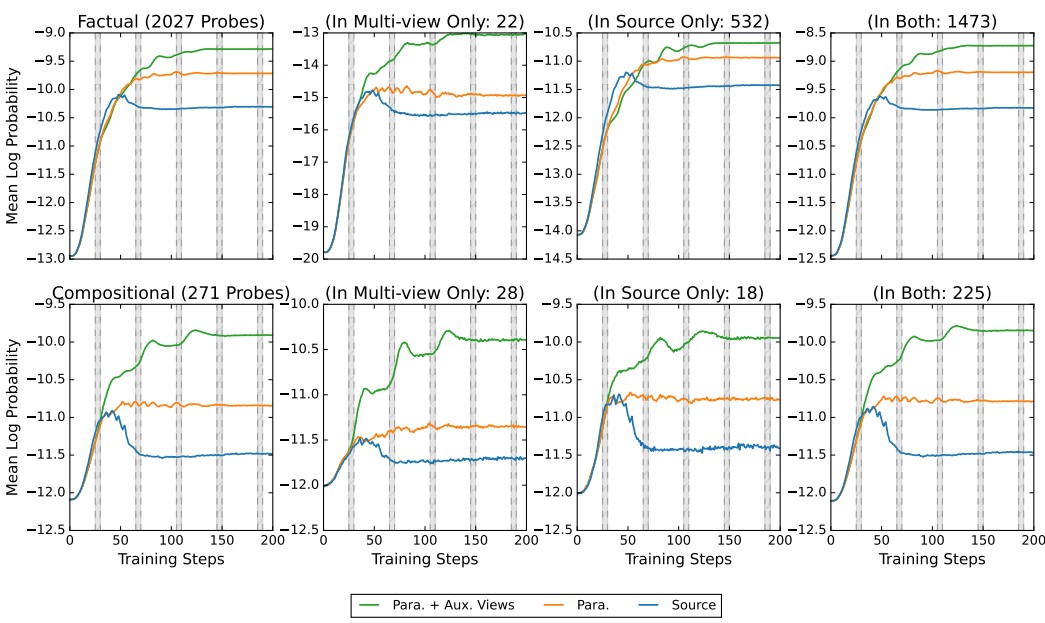

Figure 11: Stratified Performance of 7B across the three data strategies.

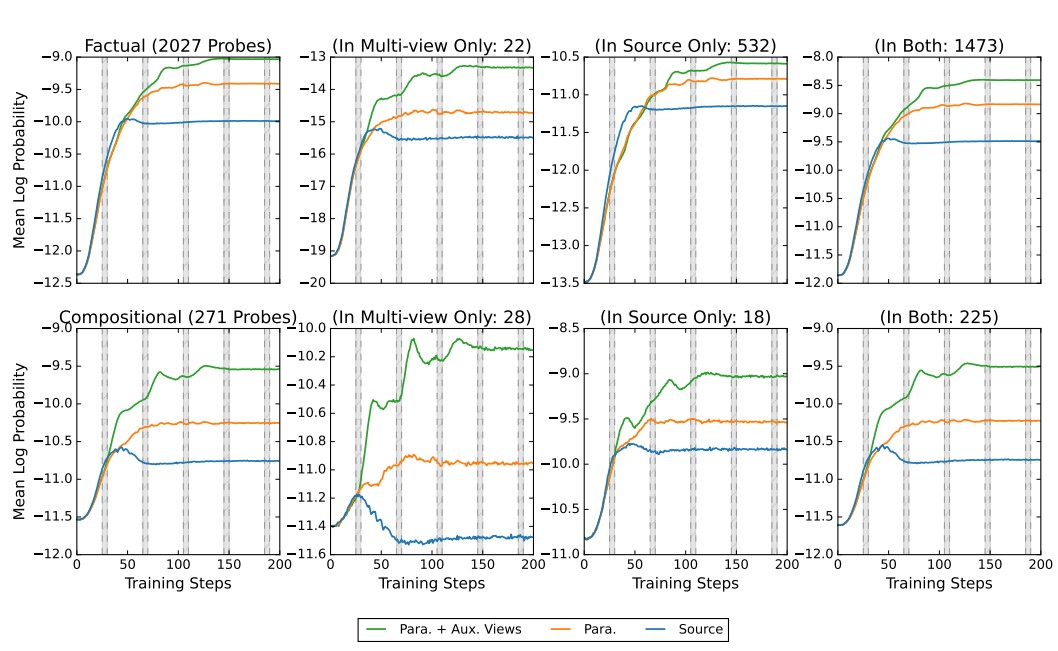

Figure 12: Stratified Performance of 13B across the three data strategies.

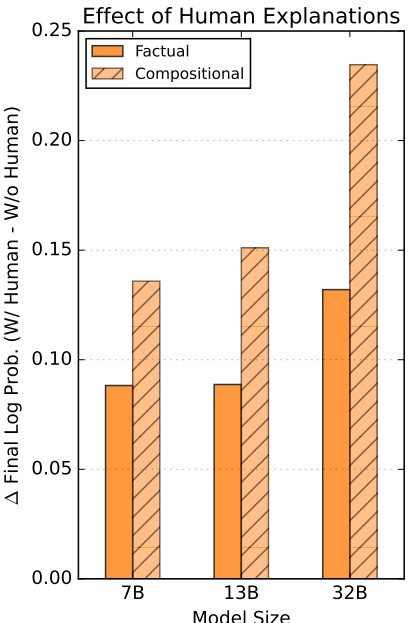

Figure 13: Performance gains from human-generated auxiliary views.

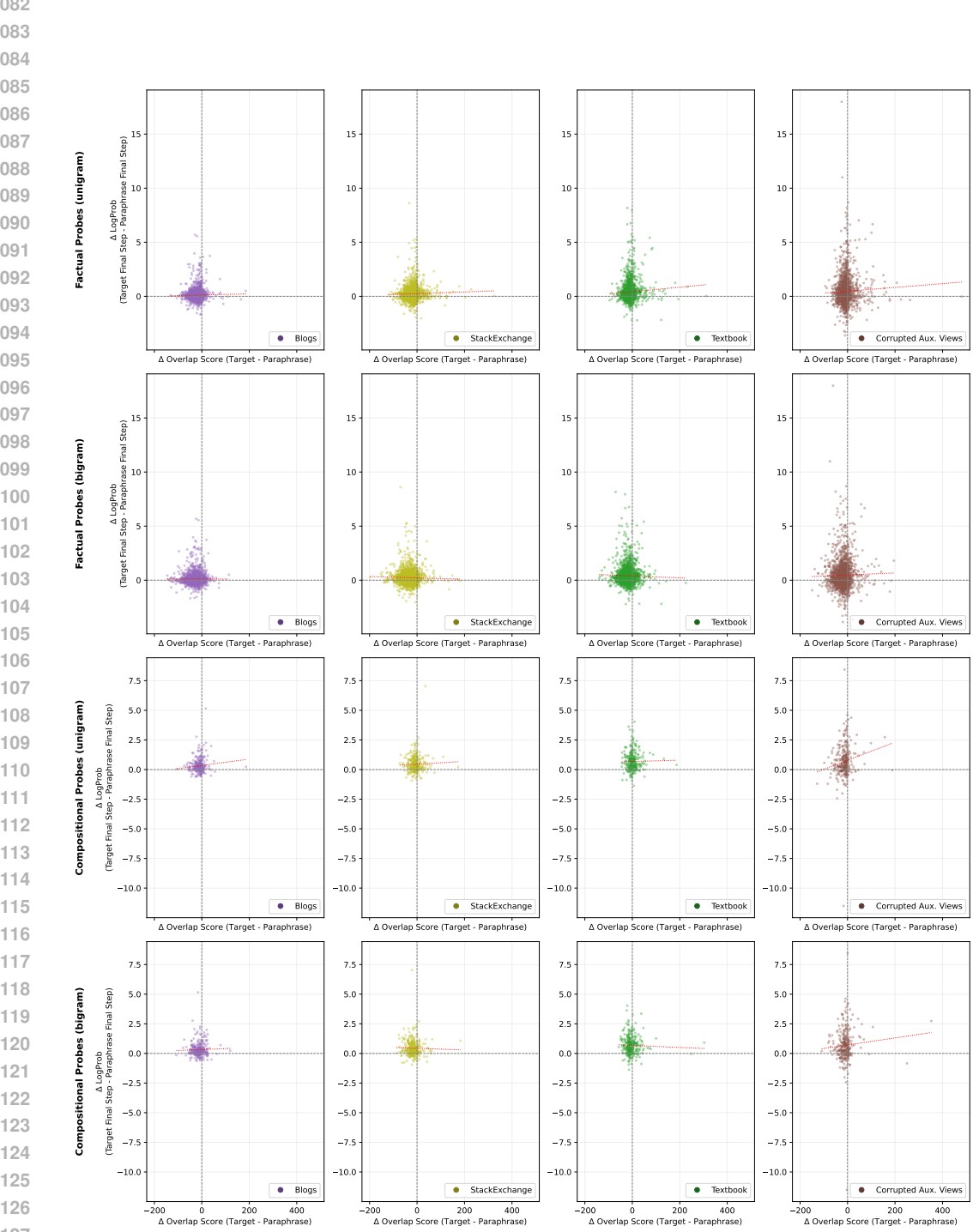

Figure 14: Analysis of lexical bias in our synthetic data.

## A.8 LIMA RESULTS

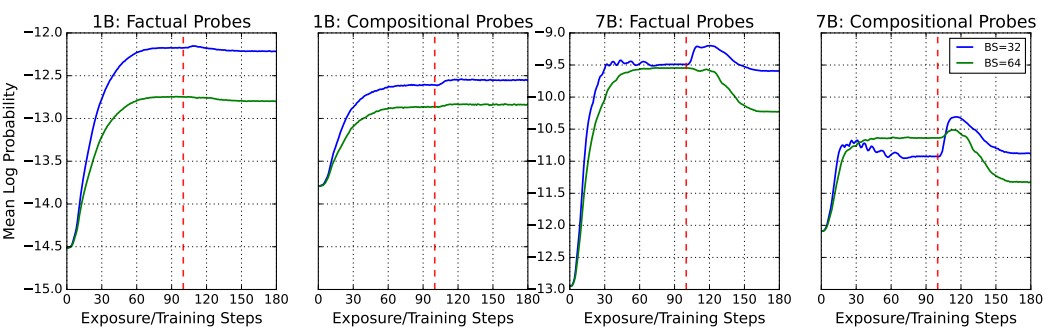

Figure 15: Comparison of batch sizes with LIMA.

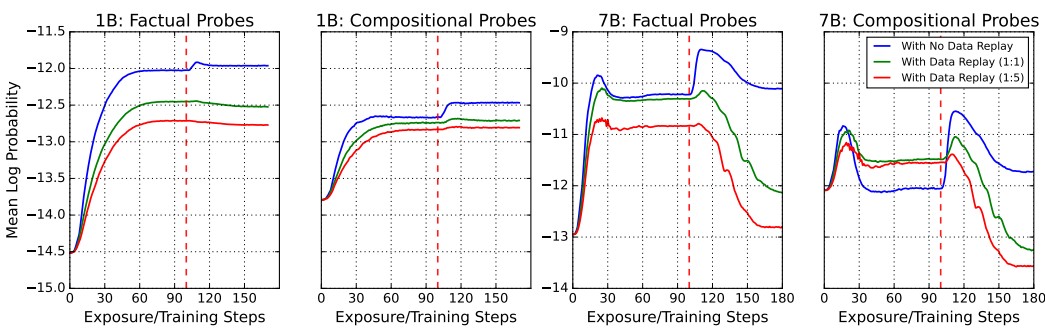

Figure 16: Comparison of data replay settings with LIMA.

## A.9 DATA REPLAY DISTRIBUTIONS

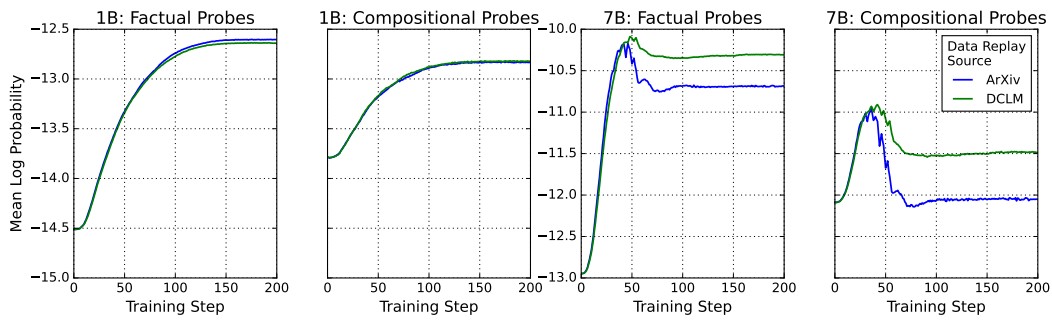

Figure 17: Arxiv vs. DCLM.

## A.10 OVERLAPPING

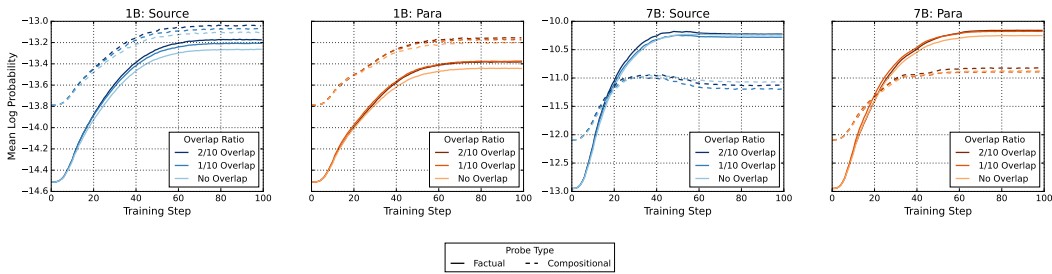

Figure 18: Comparison of overlapping ratios of chunking long documents.

## A.11 DOMAIN-LEVEL ANALYSIS

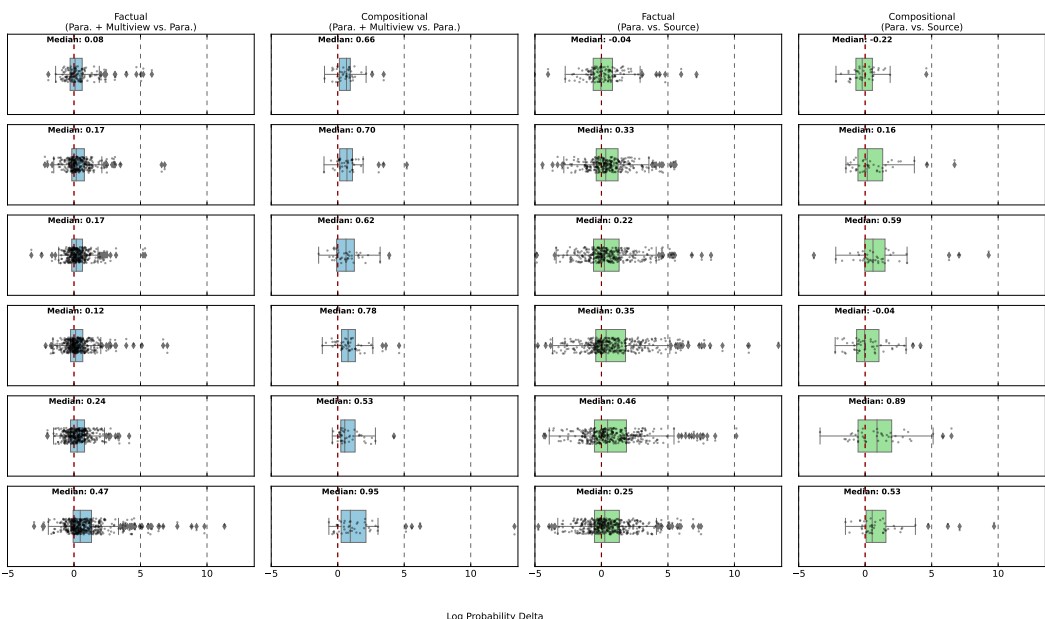

Figure 19: The advantage of auxiliary views (or multiviews in the plot) persist across all six of our papers.

## A.12 HYPERPARAMETERS FOR REPLICATION AND DETAILS FOR REPRODUCTION

Unless otherwise noted, our experiments use the following defaults: learning rate $2 \times 10^{-5}$, context size 3072, batch size 64, weight decay 0.1, cosine decay scheduler with 0.1 warm-up ratio, seed 42, max gradient norm of 1, and the AdamW optimizer ($\beta_1 = 0.9$, $\beta_2 = 0.999$, $\epsilon = 10^{-8}$) with BF16 training. For runs involving post-training with LIMA, we keep the same hyperparameters but use a 2560-token context window with packing, and following Zhou et al. (2023), train for 10 epochs. We use the TRL library to ensure no cross-attention across packed examples.

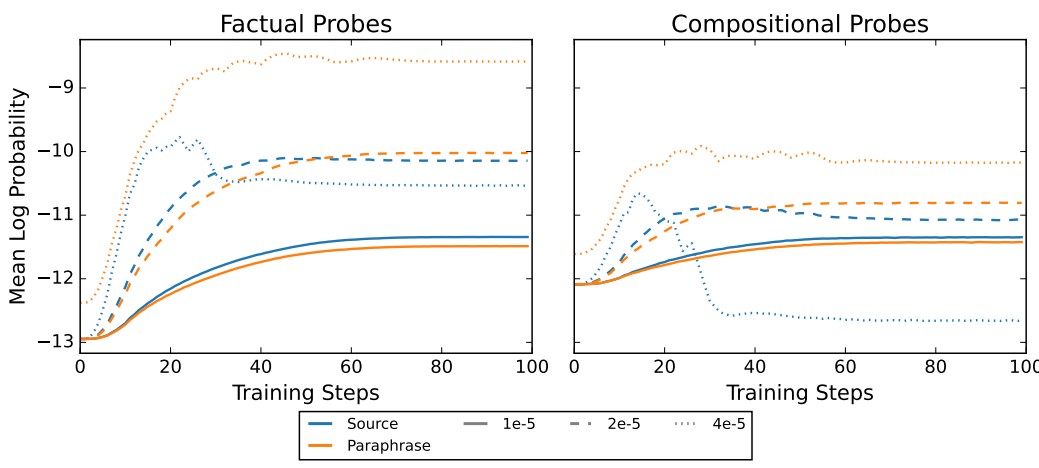

Figure 20: Training on *Source* for 500 exposures with cosine warmup and no decay. We observe an early dip followed by steady increases in target-span log-probabilities. Hits@10 improves as well, though the gains are comparatively modest and experience no initial dip.

### A.13   PROMPTS FOR PROBES

Below, we provide example prompts for different probes:

**1. Prompt to extract atomic facts from a paragraph:**

```
You will be given two inputs, a section of an academic paper for context and a single sentence
 drawn from that section. Papers often interweave various pieces of knowledge together in
academic writing. While each sentence is interwoven with others, there is atomic knowledge
that can be extracted from a particular sentence. Write questions that tests for this atomic
knowledge. Specifically, your task is to extract questions from the provided sentence with
clear answers, each 1 to 4 words long.

Extract 1-3 questions from the sentence.

### Detailed Instructions
Consider these instructions as you extract each question:
- The question should be natural and meaningful, in which the answer is considered a main fact
 presented by the sentence.
- The answer should be non-trivial and non-obvious. It should not be deducible from the
sentence itself.
- The answer to the question should be a meaningful, coherent phrase, 1-4 words long, taken
from the sentence. Simplify the answer by stripping determiners such as "some" or "a" or "an"
or "the" from the answer. Feel free to adjust the answer to fit the question, but the meaning
should be the same.
- The answer must *NOT* involve any *special characters* or *mathematical notation*. Again,
any question with an answer that contains mathematical notation should not be used.
- The question should have a a clear, single answer and *NOT* multiple valid answers.
- Each question should be written separately and independently of the other questions, so don'
t reference other questions in the same question.

### Demonstration 1
Context: "\\title{Direct Preference Optimization: Your Language Model is Secretly a Reward
Model}\n\\subsection{Can DPO scale to real preference datasets?}\nNext, we evaluate fine-
tuning performance of DPO on summarization and single-turn dialogue. For summarization,
automatic evaluation metrics such as ROUGE can be poorly correlated with human preferences~\
citep{stiennon2022learning}, and prior work has found that fine-tuning LMs using PPO on human
preferences to provide more effective summaries. We evaluate different methods by sampling
completions on the test split of TL;DR summarization dataset, and computing the average win
rate against reference completions in the test set."

Sentence: "We evaluate different methods by sampling completions on the test split of TL;DR
summarization dataset, and computing the average win rate against reference completions in the
 test set."

Questions:
- "The authors evaluate DPOs fine-tuning performance against other methods on summarization by
 sampling completions on the test split of what dataset?", Answer: "TL;DR summarization"
```

```
- "The fine-tuning performance of DPO and other methods on summarization are evaluated by
sampling completions on the test split of the TL;DR summarization dataset and computing the
average win rate against what?", Answer: "reference completions"

### Demonstration 2
Context: "\title{Direct Preference Optimization: Your Language Model is Secretly a Reward
Model}\nWhile large-scale unsupervised language models (LMs) learn broad world knowledge and
some reasoning skills, achieving precise control of their behavior is difficult due to the
completely unsupervised nature of their training. Existing methods for gaining such
steerability collect human labels of the relative quality of model generations and fine-tune
the unsupervised LM to align with these preferences, often with reinforcement learning from
human feedback (RLHF)."

Sentence: "Existing methods for gaining such steerability collect human labels of the relative
 quality of model generations and fine-tune the unsupervised LM to align with these
preferences, often with reinforcement learning from human feedback (RLHF)."

Questions:
- "What do existing methods collect to steer unsupervised language models, ?", Answer: "human
labels"
- "Existing methods for steering unsupervised language models collect human labels of the
quality of what?", Answer: "relative quality of model generations"
- "Existing methods align unsupervised language models by fine-tuning on what?", Answer: "
human preferences"
- "Existing methods for steering unsupervised language models via fine-tuning on human
preferences often use what?", Answer: "RLHF"
```

## 2. Prompt to extract atomic facts from a paragraph with context:

```
You will be given two inputs, a section of an academic paper for context, a single sentence
drawn from that section, and a question extracted from the sentence as well as its
corresponding answer. Your task is to then turn the question into a self-contained, precise
question. Approach this task step-by-step as outlined below.

While you should use your expertise on this domain to handle and understand these texts, all
information written into the questions and answers *MUST* originate from the provided context
or sentence. Do not add, infer, or correct information using your internal knowledge. Every
detail should be traceable back to the source text. As you write and rewrite the questions,
also make sure to accurately represent the knowledge in the original sentence without
distortion. Strive to use phrasing as close as possible to the original text, but prioritize
clarity and self-containment. Lastly, the questions should be written well and clearly so that
 they are easy to read.

### Instructions
The overall goal of this task is to make the questions clear by incorporating the relevant
context. This ensures the question is unambiguous and doesn't require looking back to the
source material.

For each question:
1. Rewrite the question so that it starts with one of the following templates.
    - "In the paper '{title}', ..."
    - "According to the paper '{title}',..."
    - "In the paper '{title}', the authors remark that..."
    - "In the paper '{title}', the authors state that..."
    - "According to the paper '{title}', prior work has..."
    - "In the theoretical analysis of the paper "{title}"..."
    - "In the paper '{title}', the results suggest that..."
    This is a non-exhaustive list of templates, and you should use your own judgement to choose
     the most appropriate template or modify the template to fit the sentence.
2. Add sufficient context. Specifically, use the *provided context* to supply whatever
information is needed to make the question self-contained and unambiguous. For instance, "Do
humans and GPT4 agree often with each other?" should be clarified into "In the paper '...',
did humans and GPT4 often agree or disagree with each other during the evaluation of DPO?" if
this notion was in the context of evaluating DPO in an academic paper. The goal is to ensure
someone reading just the question would understand exactly what is being asked without needing
 additional context.
3. Clarify pronouns and referential terms. Check the sentence for pronouns (it, this, that,
these, those) or demonstrative phrases (this equation, that method, these results) that refer
to entities not explicitly defined within the sentence itself. Search the surrounding context
to identify what these terms reference, then incorporate that clarifying information into the
question to make it self-contained.
4. Clarify Context-Dependent Terms. Named entities (e.g., theorems, equations, proper nouns)
do not need clarification. However, if there are unnamed or context-specific terms (e.g., $f$,
 "the model", "the loss"), clarify their full context. For instance, "the gradient" might
refer to the general concept of a gradient or to the gradient of a specific function mentioned
 earlier in the context.
5. Disambiguate experiments. There are often numerous experiments in a paper, and so supply
enough experimental context so that the question is about which experiment the question is
asking about.
```

```
6. Handle acronyms. If the answer is an acronym and the acronym appears frequently in the
context, feel free to leave it as an acronym without defining it.
7. Do not leak the answer. Please make sure that *the answer is not revealed* in the question.
 The answer should never appear in the question.
8. Maintain the essence of the original question during all of this.
9. Do not change the answer. Minor grammatical adjustments to the answer are allowed only if
necessary to fit the restructured question (e.g., adjusting verb tense, determiners like "the
").
10. Avoid quoting the source sentence directly in the question.
11. Refine Question. The rewritten question can be broken up into multiple sentences if the
question becomes verbose. Make sure the question is written clearly and grammatically correct.
 Do not put any of the context in parenthesis or followed after an "i.e.".

Think carefully and critically through this task, following the step-by-step instructions
outlined above. Then, provide the final output, listing each question and its corresponding
answer.
```

### 3. Prompt to generate comprehension (i.e. inference) questions from the text:

```
You have been given a section of an academic text. Your tasks is to test the reader's
understanding of the text. However, you should not test anything that can be recalled from
reading a single sentence. Create questions that integrate, connect, and synthesize
information across several sentences and aim at measuring a deeper understanding. Your
question must not be obvious from a single sentence already in the paper, and truly require
several sentences to synthesize the answer. Lastly, the answer to the question must be a
coherent phrase, from 1 to 5 words long.

For each question, show me the sentences in the text that you're pulling from to answer the
question. The question should be non-obvious from these sentences and require composing
information from all of them to answer the question.

Provide the output in JSON format, as a dictionary with a single key "qa_items" which is a
list of dictionaries with the following keys:
- "question": (string)
- "answer": (string)
- "text_quotes": list of sentences from the text that you're pulling from to answer the
question.
```

### A.14 PROMPTS FOR SYNTHETIC MATERIALS

### 1. Prompt to synthesize Stack Exchange style question-answer pairs:
Question generation:

```
You are a confused student reading this research paper. You are struggling with specific
concepts, details, and connections in this paper. Generate a list of several Stack Exchange
style questions that you would ask to clarify your understanding.

Your questions should:
- Vary in levels of understanding, from misled to profound.
- Vary in complexity, from simple to deep.
- Vary in type, from conceptual to detail-specific.
- Focus on clarifying the concepts and details of the paper. Do not ask tangential questions.

As you generate the questions, please make sure to consider the following:
- Make sure the questions are self-contained and unambiguous
- Please write any mathematical notation in LaTeX only e.g. "$x^2$" or "$\pi$". Do not use
unicode mathematical characters e.g. "".

For each question, provide:
- A `title` in Stack Exchange question format
- The `question_body` with context and what specifically you're confused about

## Example Question

"How can Transformers handle arbitrary length input?

The transformer, introduced in the paper Attention Is All You Need, is a popular new neural
network architecture that is commonly viewed as an alternative to recurrent neural networks,
like LSTMs and GRUs.

However, having gone through the paper, as well as several online explanations, I still have
trouble wrapping my head around how they work."

### Output Format
Provide the output as a JSON object with a single key "questions", which is a list of question
 dictionaries.
Example:
```

```
{
  "questions": [
    {
      "title": "Why does the partition function cancel out in DPO derivation?",
      "question_body": "I'm reading the DPO paper and I understand that they start with the KL-
      regularized objective, but I'm confused about how the partition function Z(x) cancels out
      when they move to pairwise preferences. Can someone explain this step intuitively?",
    }
  ]
}
```

Answer generation:

```
A graduate student has asked a question about a research paper. Provide a clear, detailed
Stack Exchange style answer that:

- Thoroughly addresses their question
- Don't make it too lengthy; it should be concise and to the point like a Stack Exchange
answer
- Write in prose rather than structured bullet points in one cohesive answer
- Provides intuitive explanations alongside technical details
- Connects to broader concepts when relevant
- Is educational and accessible

Please write any mathematical notation in LaTeX only e.g. "$x^2$" or "$\pi$". Do not use
unicode mathematical characters e.g. "". Also, please make sure that your answer is grounded
in the paper; do not provide any information that is inconsistent with the paper.

Again, please write all math in LaTeX.

Format your response as a comprehensive Stack Exchange answer.

### Example

Question:
"I know that in the math on which the transformer is based there is no restriction on the
length of input. But I still cant understand why we should fix it in the frameworks (PyTorch).
 Because of this problem Transformer-XL has been created.

Can you explain to me where this problem is hiding, please?"

Answer:
"The restriction in the maximum length of the transformer input is due to the needed amount of
 memory to compute the self-attention over it.

The amount of memory needed by the self-attention in the Transformer is quadratic on the
length of the input. This means that increasing the maximum length of the input, increases
drastically the needed memory for self-attention. The maximum length is that which makes the
model use up the whole memory of the GPU for at least one sentence (once the other elements of
 the model are also taken into account, like the embeddings which take a lot of memory).

Transformer-XL is certainly a way to take into account as much context as possible in language
 modeling (its role is analogous to truncated back-propagation through time in LSTM language
models). However, the gradients are not propagated through the attention over the memory
segment, only through the current segment.

There have been several architectural attempts to reduce the amount of memory needed by
transformers, like using locality-constraints in the attention (Dynamic Convolutions model) or
 using locality-sensitive hashing (Reformer model).

There have been other implementation attempts, like gradient checkpointing(e.g. this), which
is a general technique to run computations that don't fit at once in the GPU memory"
```

Latex formatting refinement:

```
You will be given a text. Your only task is to correct any mathematical notation inside it to
be valid LaTeX. You must not change any other part of the text.
    - Convert unicode math characters like '' to their LaTeX equivalent '$\\pi$'.
    - Ensure all mathematical expressions are enclosed in '$...$' for inline math or '$$...$$'
    for display math.
    - Return the full, corrected text.
```

## 2. Prompt to synthesize textbook style explanations:
Textbook outline generation:

```
### Instructions
You will be given a research paper and your task is to create a detailed outline for a
textbook that comprehensively explains the given research paper. But, it should go beyond mere
```

```
explaining, and be a proper pedagogical textbook that aims to fully educate the reader on
what the paper is about. The textbook should be aimed at college students who have a basic
understanding of machine learning.

The outline should:
- Break down the paper into coherent chapters.
- For each chapter, provide a:
   - title
   - description
   - list of subtopics to cover
- Cover all key concepts, methods, and results from the paper.
- Ensure a logical flow of information, from introduction to conclusion.
- While the textbook should be comprehensive, it should also articulate and to the point. Don'
t create unnecessary chapters.

### Output Format
Provide the output as a JSON object with a single key "outline", which is a list of chapter
objects. Each chapter object must have the following keys:
- "chapter_title": A string for the title of the chapter.
- "description": A string describing the chapter's content.
- "subtopics": A list of strings, where each string is a subtopic.
```

Chapter generation:

```
### Instructions
You will be given a chapter title, description, and subtopics and, based on those topics, your
 job is to write a detailed, cohesive textbook chapter addressed to a college student who is
learning this material for the first time.

The chapter should be comprehensive and suitable for someone learning this material to
understand research papers in the field. Don't just briefly describe the subtopics, but rather
 elaborate on the concepts at full length and explain them with a focus on intuition. Spell
everything out clearly so there is no ambiguity. Dedicate multiple paragraphs to each subtopic
 but be articulate and concise when appropriate. Write in full prose, rather than bullet
points. Most importantly, please make sure that your chapter is grounded in the paper; do not
provide any information or details that is not from the paper.

Start with the chapter title in the first line. Separate each subtopic with a section header
"#". Also, please write all mathematical notation in LaTeX only e.g. "$x^2$" or "$\pi$". Do
not use unicode mathematical characters e.g. "". Again, PLEASE write all math in LaTeX.
```

## 3. Prompt to synthesize blog post style explanations:

Blog post ideas generation:

```
### Instructions
You are a creative tech blogger and content strategist. Based on the provided research paper,
generate a list of a few blog posts that explain the paper in a way that is accessible to a
wider audience. They should each focus on a different, main aspect of the paper.

For each blog idea, provide:
- A `title`.
- A brief `description` of what the blog post will cover.

### Output Format
Provide the output as a JSON object with a single key "blogs", which is a list of blog objects
. Each blog object must have the following keys:
- "title": A string for the title of the blog post.
- "description": A string describing the blog post's content.
```

Blog posts generation:

```
You will be given an academic paper and a blog post idea about the paper. Write a blog post
based on the blog idea.

As you write the blog post, please make sure to consider the following:
- Write in a technical blog style. It should be less formal but not too informal. It should be
 concise and to the point.
- Simplify complex concepts from the paper for a broader audience.
- Write in full, complete sentences and prefer paragraphs over bullet points, but use bullet
points when appropriate.
- Keep all details grounded in the paper. Do not make up any information.
- Please write any mathematical notation in LaTeX only e.g. "$x^2$" or "$\pi$". Do not use
unicode mathematical characters e.g. "".

Your output should be the full text of the blog post, starting with the blog title as a
markdown header. Use '#' to denote the blog title, '##' to denote different sections, and so
on.
```

**4. Prompt to generate prior knowledge chapters:**

List of prerequisite chapters generation:

```
### Instructions
You are an expert curriculum designer. Based on the provided research paper, create a list of
textbook chapters that would provide all the necessary prior knowledge to understand this
paper. The chapters should not contain the novel ideas presented in the paper itself, but
rather the foundational concepts upon which the paper is built.

For each chapter, provide:
- A `title`.
- A general `description` of what the chapter covers.
- A list of `subtopics` that should be included.

### Output Format
Provide the output as a JSON object with a single key "chapters", which is a list of chapter
dictionaries.
Example:
{
  "chapters": [
    {
      "title": "Chapter 1: Introduction to Probability Theory",
      "description": "This chapter covers the basics of probability...",
      "subtopics": ["Random Variables", "Probability Distributions", "Bayes' Theorem"]
    }
  ]
}
```

Chapter generation:

```
### Instructions
You will be given a chapter title, description, and subtopics and, based on those topics, your
 job is to write a detailed, cohesive textbook chapter addressed to a college student who is
learning this material for the first time.

The chapter should be comprehensive and suitable for someone learning this material to
understand research papers in the field. Begin with an introduction to the chapter, then cover
 each subtopic in turn. Don't just briefly describe the subtopics, but rather elaborate on the
 concepts at full length and explain them with a focus on intuition. Spell everything out
clearly so there is no ambiguity. Dedicate multiple paragraphs to each subtopic. Write in full
 prose, rather than bullet points.

Separate each subtopic with a section header "#".

Also, please write all mathematical notation in LaTeX only e.g. "$x^2$" or "$\pi$". Do not use
 unicode mathematical characters e.g. "".
```

