# OpenReview forum: "How do Large Language Models Learn New Domain Knowledge?"
_ICLR.cc/2026/Conference — Submitted to ICLR 2026_

### Official Review · Reviewer_v3Kv · 2025-10-28

**Soundness:** 2
**Presentation:** 3
**Contribution:** 3
**Rating:** 4
**Confidence:** 4

**Summary:**

The authors study how the model acquires new knowledge in the continual pretraining process. They construct factual and compositional probes to evaluate model learning dynamics. The compositional probes is an interesting angle but the construction of the probe doesn't seems rigorous enough to support the claims in paper.

**Strengths:**

* Understanding how model learns compositional knowledge during continued pretraining is interesting and important.

* The study covers a wide ranges of model sizes.

* the ablation study and variants of training corpus are comprehensive.

**Weaknesses:**

* Claims about 49 paraphrasing reduce performance “excessive linguistic diversity may hinder factual recall” (Line 299-300), how does the author ensure the linguistic diversity?

* The diversity of the probe is not established, so the claim “This demonstrates a genuine generalization of knowledge rather than surface-level pattern matching” (Line 322-323) is not very well supported. A trivial explanation: The probes might all look similar so the trends look similar.

* Line 301 “paraphrasing enables robust knowledge acquisition over extended training”; how do the author measure “robust acquisition”?

* ``Scaffolding'' is not defined but seems important for argument of the work (i.e. appear and highlighted in abstract)

* Unclear arguments about data replay: Replay is meant to mitigate catastrophic forgetting, but learning target domain has nothing to do with catastrophic forgetting. I think the experiment results are expected. I am not sure why would training data from pretraining corpus would help learning new knowledge that doesn’t exist in pretraining corpus. I am lost about what the author want to prove.

**Questions:**

* Line 187 does "2,298 probes" include both factual and compositional probes?

* Is it correct that the compositional probe requires original + related sentence to answer?  How do the author make sure information “Related sentence” is not contained in original sentence?

* What exactly does one batch of “new knowledge” contain? Does it contain all original sentence and related sentences for a "new knowledge"? For paragraphs, prior knowledge and auxiliary views, are each of the augmentation applied to both original sentence and related sentence?

* Figure 3: why 1B with Priori knowledge has a sudden drop in mean log prob?

* Unclear why post-training is required: If the evaluation is only log prob or generate from probe prefix, why is post-training needed for the experiments? Figure 6,7,8

* Figure 10, it seems surprising that paraphrase only which has very little tokens counts (< 0.05K tokens?) are super effective. Does author have some explanation for this?

* Could the author discuss the difference between this work and prior work [1, 2] on learning dynamics?

[1] Pretrained Language Model Embryology: The Birth of ALBERT

[2] Probing Across Time: What Does RoBERTa Know and When?

---

### Official Review · Reviewer_99D6 · 2025-10-30

**Soundness:** 3
**Presentation:** 3
**Contribution:** 3
**Rating:** 4
**Confidence:** 2

**Summary:**

The paper investigates **how LLMs acquire new domain knowledge via continued pretraining (CPT)** through a series of controlled experiments. Six recent AI papers are treated as novel “domains,” and the authors construct both **factual** and **compositional** probe sets to measure learning. They compare three training strategies: (1) source-only documents, (2) paraphrasing, and (3) **auxiliary views** (textbook, blog, and StackExchange–style materials).
Key findings include:
- Learning improves with repetition but saturates after ~100 exposures.
- Paraphrasing mitigates overfitting and enhances generalization.
- **Auxiliary views act as pedagogical scaffolding**, substantially improving compositional understanding.
- **Prior-knowledge pretraining** on prerequisite concepts boosts downstream domain learning.
- **Data replay** from the pretraining corpus consistently harms new knowledge acquisition.
Experiments are conducted on OLMo2 models (1B/7B/13B), with detailed ablations and insightful discussion on the “scaffolding hypothesis” for LLM learning.

**Strengths:**

1. **Well-controlled methodology** with dual-level probes and systematic construction/validation for factual and compositional learning.
2. **Clear, interpretable insights** on paraphrasing, auxiliary views, and prior knowledge that could guide future CPT practice.
3. **Meaningful negative result:** excessive data replay hinders domain learning — a practically important observation.
4. **Thoughtful discussion** connecting findings to human learning theories (e.g., scaffolding, double descent), offering conceptual depth.

**Weaknesses:**

1. **Limited scope and scale:** Only six CS papers and models up to 13B; unclear if findings generalize to other domains or frontier-scale models.
2. **Synthetic auxiliary data:** Auxiliary “textbook/blog/Q&A” materials are LLM-generated, which might introduce lexical overlap or bias with probes despite stratified analysis.
3. **Missing baselines:** The paper does not compare against retrieval-augmented or instruction/RL-based knowledge injection methods, leaving uncertainty about whether scaffolding remains advantageous under other paradigms.

**Questions:**

1. Which components of the auxiliary bundle (textbook, blog, Q&A) contribute most to compositional gains — and do these effects persist with **human-written** materials?
2. Could alternative **replay curricula** (e.g., staged or post-cycle replay) mitigate forgetting without degrading target-domain learning?

---

### Official Review · Reviewer_ccbK · 2025-10-31

**Soundness:** 2
**Presentation:** 1
**Contribution:** 2
**Rating:** 0
**Confidence:** 4

**Summary:**

Questions: "
RQ 1: How is domain knowledge on varying levels, factual and compositional, acquired?
RQ 2: When learning new knowledge, does the gap in the LLM’s prior knowledge matter?
RQ 3: How does data replay, post-training, chunking, and other factors affect the learning of the new knowledge? "

Our corpus consists of six papers from the field of artificial intelligence

Claims:  "
(1) Acquisition of complex knowledge requires significant repetition, saturating after approximately 100 exposures in our study.
(2) Diverse, auxiliary views dramatically improve the learning of both factual and compositional knowledge in a way that paraphrasing does not.
(3) Bridging knowledge gaps by first training the LLM on prerequisite concepts significantly improves learning.
(4) Conversely, increasing the amount of data replay from the original pretraining corpus monotonically harms the acquisition of new knowledge. We also ablate several aspects of our training setup to provide pracitcal suggestions for continued pretraining. "

**Strengths:**

This paper is very interesting in its fundamental question. What is the impact of so called auxiliary views - views of the information that are intended to increase scaffolding and understanding. This is a good and important question.

**Weaknesses:**

LLMs largely lack the ability to synthesize complex, novel information from primary sources alone. - this claim is not supportable for LLMs in general. Finding that current LLMs don't do something is very different from the general class of methods lacking the ability.

When you cite a paper in the related work section, don't like 5 different papers. "Since GPT-3, the overarching narrative of LLMs
has been scale, regarding both model size and data volume (Brown et al., 2020; Kaplan et al., 2020;
Hoffmann et al., 2022; Carlini et al.; Kandpal et al., 2023; Tirumala et al., 2022)." This is not helpful and is not convincing. If they are all important, explain clearly what you intend to support with each.

Another example: "Continued pretraining has proven to non-trivial(Wang et al., 2021; Janget al., 2021; Hu et al., 2023; Ovadia et al., 2024; Hoffbauer et al., 2024; Jiang et al., 2024)." There is no purpose discernible from the paper for these citations.

Grammar and writing matter when disseminating research - this is not ready for review: "Given the recency of the focus on continued pretrainig, the exist body body is young and investigative in nature (Yıldız et al., 2024; Ou et al., 2025)"

The formal statement of the problem has an error: "What are the properties of a good training corpus K that most effectively enable the acquisition of knowledge Kfor fθ?" The corpus is C_k

"In addition to probes, we generate paraphrases, prior knowledge, and auxiliary views for each paper in the dataset, utilizing GPT-4.1 for paraphrasing and GPT-5-mini for the rest. While these alternate texts may seem to confer an implicit advantage by distilling the model’s knowledge into the training corpus, this is intentional for the auxiliary views. We treat the LLM as a proxy for domain experts who produce materials such as textbooks and blogs which enters the pretraining data. Inspired by prior works  (Gunasekar et al., 2023; Allen-Zhu & Li, 2024; Jiang et al., 2024), we focus on textbooks, Stack Exchange–style Q&A, and blogs as auxiliary views." - GPT 5 is not a domain expert in the topic of a research paper on which it has not been trained. So, the experiments can't actually evaluate the influence of auxiliary views as defined as follow on explanations by domain experts ie. the illustrated transformer as a decompression of "Attention is all you need".

monotic is not the word which is intended in figure 1. The legend in figure 1 doesn't match the actual signals in the graph. It's not clear what is being conveyed. It is very bad practice to put graphs on the same axis with different values on that axis. It makes the results incomparable.

While the central question of the paper is good, the presentation in the paper makes it difficult to judge the voracity of the evidence. The presentation of the paper is not ready for publication.

**Questions:**

No questions.

---

### Official Review · Reviewer_PDMR · 2025-11-01

**Soundness:** 3
**Presentation:** 4
**Contribution:** 3
**Rating:** 6
**Confidence:** 4

**Summary:**

This work analyzes the knowledge acquisition dynamics of LLMs through the controlled setup of continued pretraining. The analysis reveals that the increase of log probability induced by repeated exposure leads to saturation, while providing the model with multiple views of given knowledge improves its learning and generalization. Based on the extensive analysis, the paper conjectures that LLMs largely lack the ability to infer novel compositions of primary knowledge sources without the aid of auxiliary views.

**Strengths:**

(S1) The experiments are appropriately designed to deal with each research question.

(S2) The results reveal several interesting phenomena, in particular the double-descent-like behavior and the effect of model size on the benefit of auxiliary views.

(S3) The paper is well-written with clean and precise language.

**Weaknesses:**

(W1) While providing various insights, (as mentioned in the discussion) the experiment relies on a single domain of research paper understanding, and it is unclear whether the same result can be applied to other specific domain knowledge, for example, structured documents that require the expertise in medical or legal domain.

(W2) The mechanistic understanding of the observed behaviors is limited, making the claims on the ‘structure’ stay hypothetical (for example, “auxiliary views help the model build a more structured knowledge representation”, in L345). It would be great to see additional analysis of what makes it different from the model trained with/without auxiliary views, in terms of the information encoded in representations or gradients. In addition, studies on synthetic data reveals that grokking often accompanies certain structures of representation (e.g., [1]). Will such “grokking-like” behavior observed in this study related to some induced structure inside the parameters?

(W3) It has been actively discussed in recent studies that domain adaptation often leads to catastrophic forgetting. While the current analysis reveals many potential advantages of domain knowledge acquisition under certain conditions, it might be at the cost of forgetting unrelated knowledge that should be maintained. Could you share your thoughts on the forgetting dynamics under domain adaptation, or additional experiments to rule out that providing auxiliary views does not aggravate catastrophic forgetting?


[1] https://arxiv.org/abs/2405.15071

**Questions:**

Please refer to the points in the weaknesses section.

---

### Author Response · Authors · 2025-12-03
**Revisions and Overall Rebuttal**

## Rebuttal to Reviewers' Feedback and Manuscript Improvements

We sincerely thank the reviewers for their thoughtful reviews and constructive feedback. We acknowledge the points raised and provide rebuttals below. Additionally, we have significantly refined our manuscript to address these concerns. To facilitate the review process, we provide a summary of the changes and a reiteration of our motivation.

***

### Motivation Reiteration: The Significance of Natural Language Formulation

There remains a significant mystery surrounding **pre-training**. Due to the massive scale involved, it is often unclear exactly what is occurring within the model or what capabilities should be expected given the opacity of the data (Ye et al. 2023).

In particular, we find that the significance of the **natural language representation of knowledge** is understudied. Allen-Zhu et al. (2024) observe that augmenting Question-Answering (QAs) for a fraction (p) of facts increases memorization for other facts (1-p). This suggests the existence of **specific natural language formulations enable superior encoding** during pre-training, and this can affect more than just the original knowledge. While pre-training typically bypasses these concerns by massively sampling from the human writing distribution, this diverse coverage of natural language is not guaranteed for long-tail information and future information.  Our work thus investigates the significance of **natural language formulation**, specifically testing the impact of **"auxiliary views,"** which we believe are common in human data. We essentially ask:

> “Can an LLM learn calculus simply by reading lecture notes, or does it require a teacher to rephrase concepts, work through problems, and spell details out?”

***

### Overview of Manuscript Improvements

* The manuscript—including the **Introduction, Related Works, and Problem Formulation**—has been revised to better reflect the motivation summarized above.
* We have improved the presentation and clarity of **all figures and captions**.
* We have conducted **new experiments** to bolster our results and control for possible confounders.

***

### Overview of New Results

We present three main new results:

#### 1. Reconciling Paraphrasing Efficacy

We identify a predictable setting in which the **efficacy of paraphrasing decreases**. This finding helps explain the conflicting results between Allen-Zhu et al. (2023) and Chang et al. (2024) regarding the utility of paraphrasing.

#### 2. Mitigating Lexical Bias and Leakage

We have implemented rigorous controls to prevent **lexical bias or "cheating"** in the synthetic tests. Initially, we stratified probes to ensure answers appeared only in the original paper. To address concerns that the probe text itself (not just the answer) might have leaked into the synthetic text, we added two additional safeguards:

* **Adapted BM25 Metric:** We created a new metric based on **BM25** to measure the overlap between probes and synthetic text, weighted by the text's appearance in the original papers, to quantify any potential "advantage."
* **Lexical Ablation:** We **ablated our auxiliary views** by removing all words present in the probes, replacing 11.1% of the words with the **EOS token**. The performance benefits remained statistically significant, suggesting the model is not relying on simple lexical overlap.

#### 3. Comparison to Human Texts

Based on feedback, we tested how **human texts compare**. While the sample size was limited to two domains, we found that the **effect is present and grows with model size**. This suggests that **larger models increasingly benefit from high-quality explanations** and "understanding" rather than just rote pattern matching.

Ye, Qinyuan, et al. "How predictable are large language model capabilities? a case study on big-bench." Findings of the Association for Computational Linguistics: EMNLP 2023. 2023.

Chang, Hoyeon, et al. "How do large language models acquire factual knowledge during pretraining?." Advances in neural information processing systems 37 (2024): 60626-60668.

Allen-Zhu, Zeyuan, and Yuanzhi Li. "Physics of language models: Part 3.1, knowledge storage and extraction." arXiv preprint arXiv:2309.14316 (2023).

---

### Meta-Review · Area_Chair_QDw5 · 2025-12-22

**Summary:**

There is consensus that the paper addresses an important  question: how large language models acquire complex and compositional knowledge during continued pretraining. The argument behind this work is that one can more reliably improve knowledge retention by adding in auxiliary views. The experimental design is viewed as systematic with alignment between research questions and empirical setup. Reviewers liked the use of probes (factual and compositional), and ablations (paraphrasing rates, auxiliary views, prior knowledge, replay).

That said, multiple reviewers question whether conclusions drawn from a small set of CS research papers generalize to other domains, particularly high-stakes or highly structured domains such as medicine or law, or to frontier-scale models. The mechanistic understanding of why auxiliary views work remains limited. Specifically there is little effort expended in understanding e.g. through gradients, activation monitoring, mechanisms on how auxiliary views change learning dynamics. Currently the paper only uses downstream performance as a proxy for quality of improvement. Several reviewers also note missing or weak baselines (e.g., retrieval-augmented or instruction-based methods), unclear definitions (e.g., scaffolding, robustness), ambiguous probe diversity, and confusion around the motivation and interpretation of replay experiments and post-training. Finally one review felt the paper has a lot of limitations around presentation quality: imprecise claims about LLM capabilities, citation overloading without clear purpose, grammatical errors, unclear figures, and technical mistakes in problem formulation, all of which make it difficult to cleanly weigh the evidence.

**Reviewer Concerns:**

While I think some of the clarity and grammar issues may have been addressable, the additional baselines would improve the work. In addition to what the reveiwers point out there is also work on incorporating auxiliary context during training (https://arxiv.org/abs/2503.09032, https://arxiv.org/abs/2503.01821) that would also serve as reasonable baselines.

For me, the biggest decision point was the Weakness 2,3  by reviewer PDMR in terms of better elucidating mechanisms on how this style of learning actually operates would improve the work as well as more clarity around whether forgetting is an issue.

**Reviewer Scores:**

Given the response I am not convinced any of the reviewers would have changed their score since the response did not cleanly breakdown which specific weakness was being responded to and several weaknesses remain.

---

### Decision · Program_Chairs · 2026-01-26

Reject